# Glucocorticoid Receptor Isoforms in Breast Cancer Raise Implications for Personalised Supportive Therapies

**DOI:** 10.3390/ijms252111813

**Published:** 2024-11-03

**Authors:** Henriett Butz, Viktória Vereczki, Barna Budai, Gábor Rubovszky, Rebeka Gyebrovszki, Ramóna Vida, Erika Szőcs, Bence Gerecs, Andrea Kohánka, Erika Tóth, István Likó, Imre Kacskovics, Attila Patócs

**Affiliations:** 1Department of Molecular Genetics, The National Tumour Biology Laboratory, Comprehensive Cancer Centre, National Institute of Oncology, 1122 Budapest, Hungary; vereczki.viktoria@oncol.hu (V.V.); budai.barna@oncol.hu (B.B.); patocs.attila@oncol.hu (A.P.); 2Department of Oncology Biobank, National Institute of Oncology, Comprehensive Cancer Centre, 1122 Budapest, Hungary; 3HUN-REN-SU Hereditary Tumours Research Group, Hungarian Research Network, 1089 Budapest, Hungary; 4Department of Laboratory Medicine, Semmelweis University, 1085 Budapest, Hungary; 5Department of Anatomy, Histology and Embryology, Semmelweis University, 1085 Budapest, Hungary; 6Department of Thoracic and Abdominal Tumours and Clinical Pharmacology, National Institute of Oncology, Comprehensive Cancer Centre, 1122 Budapest, Hungary; rubovszky.gabor@oncol.hu; 7Department of Surgical and Molecular Pathology and the National Tumour Biology Laboratory, Comprehensive Cancer Centre, National Institute of Oncology, 1122 Budapest, Hungary; 8Department of Immunology, Institute of Biology, Eötvös Loránd University, 1053 Budapest, Hungary; imre.kacskovics@ttk.elte.hu; 9ImmunoGenes-ABS Ltd., 2092 Budakeszi, Hungary

**Keywords:** glucocorticoid receptor, breast cancer, triple-negative breast cancer, glucocorticoid receptor β, immunohistochemistry, survival, outcome

## Abstract

Glucocorticoid receptor (GR) activation may promote metastasis in oestrogen receptor-negative and triple-negative breast cancer (TNBC). However, the role of the GRβ isoform, which has opposing effects to the main isoform, has not been studied in clinical samples. We aimed to analyse the intracellular localisation of total GR and GRβ in vitro using plasmid constructs and fluorescent immunocytochemistry. Additionally, our goal was to perform immunostaining for total GR and GRβ on two cohorts: (i) on 194 clinical breast cancer samples to compare the expression in different molecular subtypes, and (ii) on 161 TNBC samples to analyse the association of GR with survival. We supplemented our analysis with RNA data from 1097 TNBC cases. We found that in the absence of the ligand, GR resided in the cytoplasm of breast cancer cells, while upon ligand activation, it translocated to the nucleus. A negative correlation was found between cytoplasmic GRtotal and Ki67 in luminal A tumours, while the opposite trend was observed in TNBC samples. Tumours with strong lymphoid infiltration showed higher cytoplasmic GRtotal staining compared to those with weaker infiltration. Patients with high nuclear GRtotal staining had shorter progression-free survival in univariate analysis. High cytoplasmic GRβ was a marker for better overall survival in multivariate analysis (10-year overall survival HR [95% CI]: 0.46 [0.22–0.95], *p* = 0.036). As a conclusions, this study is the first to investigate GRβ expression in breast tumours. Different expression and cellular localisation of GRtotal and GRβ were observed in the context of molecular subtypes, underscoring the complex role of GR in breast cancer. An inverse association between cytoplasmic GRtotal and the Ki67 proliferation index was observed in luminal A and TNBC. Regarding the impact of GR on outcomes in TNBC patients, while cytoplasmic GRβ was associated with a better prognosis, patients with nuclear GRtotal staining may be at a higher risk of disease progression, as it negatively affects survival. Caution should be exercised when using glucocorticoids in patients with nuclear GR staining, as it may negatively impact survival.

## 1. Introduction

Recent translational studies indicate that glucocorticoid receptor (GR) activation increases tumour heterogeneity and metastasis formation, suggesting caution in using glucocorticoids along with chemotherapy for breast cancer patients [1,2,3,4]. In preclinical studies, increased glucocorticoid receptor activity was demonstrated in triple-negative, metastatic breast cancer that led to increased colonisation on metastatic sites and reduced survival [2]. Chemotherapy (based on anthracyclines and taxanes) is the standard treatment strategy for metastatic triple-negative breast cancer (TNBC) [5]. Glucocorticoids are routinely administered as supportive therapy to counteract adverse symptoms associated with chemotherapy and radiation therapy [6,7]. Dexamethasone ameliorates chemotherapy-induced cytotoxicity and prevents allergic reactions [6,7]. Additionally, glucocorticoids inhibit anti-tumour immune responses and are linked to increased tumour cell survival [7] that could potentially influence the effect of immunotherapy as well.

First, on the RNA level, Pan et al. demonstrated that high GR expression was associated with worse survival in TNBC [8]. However, immunohistochemistry studies have been somewhat controversial and no subtype-specific data have been reported despite the observation of RNA-based studies that in oestrogen receptor-positive (ER+) breast cancer, high GR gene expression correlated with good outcome [9,10,11,12,13,14,15].

GR, a nuclear receptor, is located in the cytoplasm in its unliganded form. Upon ligand binding, GR undergoes a conformational change, translocates to the nucleus and modifies gene expression [16,17]. Additionally, GR has nongenomic effects in the cytoplasm, such as activating the phosphoinositide 3-kinase/protein kinase B pathway and NF-κB, regulating JNK function, inducing apoptosis, and affecting metabolic pathways through mitochondrial and membrane-bound GR [16,17].

Immunohistochemical (IHC) studies of normal breast tissue show GR positivity in both cytoplasm and nuclei [18,19]. Furthermore, Conde et al. have reported that during breast cancer development, GR shifts from the nucleus to the cytoplasm [20].

Several GR isoforms, including GRα and GRβ, diversify GR function. GRβ, an inhibitor of GRα, is produced through alternative splicing, affecting its ligand-binding domain [17,21]. It is described that GRβ is expressed in tissues at lower levels compared to GRα or not expressed until a disease state, such as cancer, or prolonged glucocorticoid treatment occurs [21]. It is suggested that the subcellular localisation of GRβ depends on cell type [21,22]. In breast tissue, we previously demonstrated the presence of GRβ in both normal and cancerous breast specimens mostly in cytoplasmic localisation in tumours, while some samples also showed nuclear positivity [19]. Within the cell nucleus, GRβ competes with GRα for binding to glucocorticoid response elements (GREs), forms heterodimers with GRα, interacts with proteins that prevent GRα activity, recruits deacetylase complexes, and has GRα-independent transcriptional activity [21]. However, the cytoplasmic and nuclear function of GRβ in breast cancer remain controversial [21].

Previous IHC reports have used antibodies that detect only or predominantly nuclear GR, regardless of isoform composition [9,10,11,12,13,14,15,23,24]. Therefore, we are lacking data on GRβ’s clinicopathological relevance due to difficulties in discriminating GRβ from the other GR isoforms.

Therefore, in this study, using a specific and validated antibody [19] against GRβ, we aimed to analyse the intracellular localisation of the total GR (GRtotal) and the β isoform of the GR (GRβ) in vitro using transfection of plasmid constructs encoding the GR isoforms and confocal fluorescent immunocytochemistry. As a second objective, we performed immunostaining for GRtotal and GRβ on two clinical breast cancer cohorts: (1) one cohort that consisted of 194 clinical breast cancer samples to compare different molecular subtypes, and (2) a cohort of 161 TNBC samples to evaluate the association of GRtotal and GRβ on survival.

## 2. Results

### 2.1. GR Cellular Localisation and Staining Characteristics

In the human breast cancer specimens, we observed both cytoplasmic and nuclear staining (Figure 1A,B) in the cohorts investigated (Table 1). As cytoplasmic staining is often considered non-specific in routine pathology, we analysed the intracellular localisation of GRtotal and GRβ in vitro by plasmid constructs and fluorescent immunocytochemistry. To support the cytoplasmic presence of GR, we demonstrated that in the absence of GR agonist treatment, GR predominantly resided in the cytoplasm and translocated to the nucleus upon activation (Figure 1C). Therefore, we identified cytoplasmic staining as a specific signal and we analysed the cytoplasmic and nuclear staining separately during the subsequent IHC evaluations.

In human breast cancer specimens, both cytoplasmic and nuclear staining were observed (Figure 2), and samples with positive and negative staining were identified. Positive cytoplasmic GRtotal (GRt-cpl), GRβ (GRβ-cpl), nuclear GRtotal (GRt-nucl), and GRβ (GRβ-nucl) were detected in 85%, 80%, 26%, and 7% of the samples, respectively, while the rest of the samples were negative. A significant correlation among GR staining parameters with each other and with ER, PR, and Her2 (Table 2) was observed.

Regarding the staining strength of positive samples, an overall weak-to-moderate cytoplasmic GRtotal (GRt-cpl) and GRβ (GRβ-cpl) staining were detected, accompanied by nuclear positivity in a small proportion (26 and 7%, respectively) of the samples. We observed that overall, a small percentage of samples showed high GR expression: GRt-cpl: 15.6% (25/160), GRt-nucl: 5% (8/160), GRβ-cpl: 47% (74/158), and GRβ-nucl: 5% (8/158). As samples with high GRt-cpl staining were relatively few, we validated the distribution of GRtotal expression at the RNA level in an independent cohort of 1097 TNBC patients. We found that the proportion of patients with high GR levels (strong GRt-cpl staining vs. high GR gene—*NR3C1*—expression) was similar (15.6% vs. 16.3%, *p* = 1.000) (Figure 1D,E).

### 2.2. Expression of GRtotal and GRβ Isoform in Breast Cancer Intrinsic Molecular and Main Histological Subtypes

Cytoplasmic staining of GRtotal was higher in ER+ luminal A (LumA) and luminal B (LumB)-Her2+ samples compared to TNBC samples (Figure 3A). GRβ-cpl staining was the strongest in LumB-Her2-positive samples (Figure 3B). No differences in nuclear GR staining were observed among the intrinsic molecular subtypes.

Comparing ductal and lobular breast cancer, both GRt-cpl and GRβ-cpl staining were higher in lobular cancers (Figure 3C,D).

Based on the opposite function of GR isoforms among different breast cancer subtypes, the associations of GRtotal and GRβ with clinicopathological parameters were analysed separately in LumA and TNBC. In LumA samples, GRt-cpl staining negatively correlated with Ki67 (Table 3), supported by weaker GRt-cpl cytoplasmic staining in LumB-Her2-negative samples compared to LumA, which only differs in their Ki67 indices (Figure 3A).

### 2.3. Correlation of GR Staining with Clinicopathological Parameters in TNBC

Based on the clinical GR relevance in TNBC, we used an independent cohort with a larger sample size and with available long follow-up data (n = 161). Compared to LumA samples, in TNBC, both GRt-cpl and GRβ-cpl correlated positively with proliferation marker Ki67 (Table 3).

Tumours with prominent lymphocytic infiltrate (≥50% stromal lymphocytes) showed higher GRt-cpl staining than those with weaker infiltration (Figure 4A). Higher GRβ-nucl staining was linked to more extensive vascular infiltration (Figure 4B).

In survival analysis, GRt-cpl and GRβ-cpl staining showed opposite trends. Tumours with higher GRt-cpl staining had shorter progression-free survival (PFS) and overall survival (OS), while high GRβ-cpl staining did not associate with survival (Figure 5A–D). GRt-nucl staining was associated with worse survival (Figure 5B).

In univariate analysis, besides GRt-nucl, we found among others, age, vascular and perinodal invasion, Nottingham prognostic index, tumour size, stage, and positive lymph node ratio as prognostic factors (Table 4). After excluding multicollinearity, multivariate regression indicated that besides positive lymph node ratio and grade, GRβ-cpl showed an association with better OS (Table 5), while no significant association with PFS was observed.

## 3. Discussion

Previously, only nuclear GR staining was evaluated in relation to clinicopathological parameters in breast cancer [9,10,11,12,13,14,15,23,24]. However, the presence of cytoplasmic GR has important non-transcriptomic functions in the breast, and a shift from nuclear to cytoplasmic GR has been noted during breast cancer development [17,18,19,20,25]. Consistent with this, we detected GRt-cpl and GRβ-cpl in 85% and 80% of all breast cancer samples, respectively, indicating a high prevalence and potential significance. Using a different antibody, Al-Alem observed cytoplasmic GR in 43% of samples [10]. Among the human breast cancer specimens, we found overall weak-to-moderate cytoplasmic GRtotal (GRt-cpl) and GRβ (GRβ-cpl) staining. We verified the small proportion of high GRt-cpl samples (16.3%) with an independent set of over 1000 breast cancer samples and found a similar rate.

The literature reports on nuclear GR expression are heterogeneous, ranging from a consistent lack of GR staining in almost all invasive breast carcinomas (except metaplastic tumours) to varying positivity rates between 44% and 83% [9,10,12,24]. In our study, nuclear GR positivity was less frequent, with 26% and 7% for GRtotal and GRβ isoform, respectively. Additionally, GR staining parameters were not independent of each other; positive correlations were observed between GRt-cpl and GRβ-cpl, and between GRt-cpl and GRt-nucl [26]. The correlation of GR presence with clinicopathological parameters in the literature is also highly controversial. Some authors found no correlation between GR and breast cancer subtypes or characteristics [10]. However, in line with other reports, we found a fairly consistent positive correlation between GRt-cpl and ER [9]. GR expression was higher in ER+ breast cancer compared to ER- tumours and showed low expression in TNBC [9,23]. Similarly to our results, higher GR staining has been observed in lobular compared to ductal breast tumours [10].

Based on literature data, the function of GR depends on the presence of ER [27], so we investigated the correlation of GR staining in LumA and TNBC samples. In ER+ samples, in accordance with others [9], we found that GRt-cpl negatively correlated with Ki67, supported by the comparisons of LumA and LumB-Her2- samples. In TNBC, however, GRt-cpl and GRβ-cpl showed an opposite trend. Indeed, it has been shown that activated GR binds to and represses the enhancer regions of the ER-mediated cell cycle genes’ (e.g., *CCND1*, *CDK2*, and *CDK6*) in ER+ breast cancer cells [28]. The interaction between ER and GR was reported to have a clinical impact, as in ER+ breast cancer low GR expression associated with worse outcomes and high Ki67 [8,9]. TNBC samples with strong lymphoid infiltration had higher GRt-cpl, while those with extensive vascular infiltration had higher GRβ-cpl. This phenomenon requires further research, as GR generally has an immune-suppressive nature and GRβ has been linked to glucocorticoid resistance [29]. These findings could impact tumour-immune surveillance and influence immune-checkpoint inhibitor treatment efficacy in TNBC patients [30,31], but these hypothesis needs further studies.

In our univariate analysis, high GRt-nucl staining associated with worse progression-free survival, but this significance was lost in multivariate analysis. This aligns with Abduljabbar et al., who found a similar pattern for breast-cancer-specific survival [9].

Clarifying the role of GRβ in breast cancer is a high priority due to its suggested antagonistic effect compared to the main GR isoform GRα. However, there is limited information on GR splice variants in clinical breast cancer specimens [21,32,33,34,35,36,37]. To our knowledge, no immunohistochemistry studies on the GRβ isoform in breast cancer have been published. The sequence similarity between GRα and GRβ makes specific discrimination challenging [19]. Therefore, we developed and validated an antibody against GRβ [19]. Using this antibody, we observed an opposite trend in association of GRβ-cpl with survival compared to cytoplasmic GRtotal. In multivariate analysis, samples with high GRβ-cpl staining showed significantly longer OS compared to low GRβ-cpl samples. This may be due to the different transcriptional and cytoplasmic actions of the GRβ isoform [21]. While GRα and GRβ can influence each other’s activity, GRβ also has intrinsic, GRα-independent transcriptional activities [38,39]. Indeed, in breast cancer, we observed different effects of GRα and GRβ on proliferation, migration, and apoptosis in the context of ER [19].

Several studies have reported GR’s non-genomic, cytoplasmic effects through protein–protein interactions and post-translational modifications [40]. In our cohorts, GRβ showed lower nuclear presence compared to GRtotal. Though, the cytoplasmic function of GRβ in breast cancer cells remains unknown.

Although cytoplasmic staining is challenging to assess in clinical samples, the amount of GRβ-cpl may serve as a useful marker to identify breast cancer patients with different outcomes.

## 4. Materials and Methods

### 4.1. Samples, Pathological Characterisation, and Tissue Microarray Construction

We prospectively collected 194 samples in the first cohort to evaluate the expression of GR isoforms across molecular subtypes of treatment-naïve breast cancer (Table 1, Appendix A). Formalin-fixed, paraffin-embedded (FFPE) blocks not older than 2 years were selected to ensure proper staining optimisation and preanalytics. As a second independent cohort, investigating the effect of GR on prognosis and outcome, 161 TNBC cases were selected, having 10–16-year follow-up data available.

Pathological characterisation was conducted as part of the routine diagnostics, and breast cancer subtype classification was conducted according to current guidelines [41,42,43] using immunohistochemistry (Appendix A). Lymphocytic infiltrate was quantified according to recommendations by the International TILs Working Group 2014 (lymphocyte infiltration no: 0%, yes: 1–49%, and strong: ≥50%) [44].

Based on the pathological characterisation, haematoxylin and eosin (H&E)-stained sections from each FFPE block were used to define the tissue microarray (TMA) diagnostic areas. Depending on the size of the tumour in the block, two to three random representative cores with a 1.5 mm diameter were obtained and inserted into a recipient paraffin block using a tissue arrayer (3DHISTECH, TMA Master II, Budapest, Hungary).

*NR3C1* gene expression of 1097 TNBC sample RNA-seq data was retrieved from a total of 11,688 samples from the Genotype-Tissue Expression (GTEx) portal (version no. 7–15 May 2019) through TNMplot.com [45].

The study was approved by the Scientific and Research Committee of the Medical Research Council of the Ministry of Health, Hungary (BMEÜ/1774-1/2022/EKU).

### 4.2. Immunohistochemistry Staining and Scoring

For immunohistochemistry analysis, FFPE sections were processed as described previously [19]. For total GR and GRβ staining, a polyclonal antibody against human GR N-terminal (GTX101120, 1 mg/mL; GeneTex, Irvine, CA, USA) was applied at a dilution of 1:100, and a mouse monoclonal antibody produced and characterised by ImmunoGenes Ltd., Budakeszi, Hungary (10G8; 0.94 mg/mL), which was used at a dilution of 1:4000, as previously described [19]. Biotinylated goat anti-mouse antibody (BA-9200 Vector Laboratories, Burlingame, CA, USA) or anti-rabbit (BA-1000, Vector Laboratories, Burlingame, CA, USA) were applied as secondary antibodies with DAB solution as the chromogen (Vector Laboratories, Burlingame, CA, USA, Impact^®^ DAB Substrate, Peroxidase (HRP) [19]. Sections were counterstained with haematoxylin (Novolink, Leica Biosystems Newcastle Ltd., Newcastle Upon Tyne, UK).

Staining patterns and intensities were assessed by two of the three experts in histology (V.V., H.B. and R.Gy.) and cross-referenced by an expert pathologist (E.T.). Interobserver variation was assessed by Krippendorff’s alpha (ratio) [46]: 0.7615 for cytoplasmic GR and 0.759 for nuclear GR indicated acceptable agreement.

For cytoplasmic staining, scores were given from 0 to 3+ determined by a scale from negative to the strongest positivity (Figure 1A). For nuclear positivity, the percentage of positive cells was given by counting 100 tumour cells (Figure 1B). Final scores were the average of the scores given by two investigators. Cytoplasmic and nuclear scores were examined separately during statistical evaluation. GR staining score cut-offs were determined as: GRt-cpl high: ≥2, low: <2; GRt-nucl high: ≥65, low: <65; GRβ-cpl high: ≥0.83, low: <0.83; GRβ-nucl high: ≥30, low: <30.

### 4.3. Immunocytochemistry Following Transfection of Expression Vectors Encoding GR for Evaluation of Cellular Localisation

For the assessment of cellular localisation of GR, in vitro cell culture experiments and treatment were performed, as previously reported [19]. For in vitro transfection, HeLa cells, a routinely used and relevant model, were chosen to study GR’s cellular localisation due to their epithelial origin and glucocorticoid receptor expression [39,47,48,49,50,51]. Cells were cultivated in MEM (#31095029, Gibco, Thermofisher Scientific, Waltham, MA, USA) and supplemented with 10% fetal bovine serum (#10270106, Gibco, Thermofisher Scientific, Waltham, MA, USA), 1% sodium-pyruvate (#11360070, Gibco, Thermofisher Scientific, Waltham, MA, USA), and 1% antibiotic-antimycotic solution (Sigma-Aldrich, Merck, Kenilworth, NJ, USA) in a humidified incubator infused with 5% CO2 at 37 °C. Before treatment, cells were grown in their complete media using hormone-free fetal bovine serum (FBS) for 48 h. Hormone-free FBS was prepared by incubating and mixing 0.1 g dextran-coated active charcoal (#C6241, Sigma-Aldrich, Merck, Kenilworth, NJ, USA) per 6 mL FBS for 24 h at 4 °C. After 24 h, mixtures were centrifuged 300× *g* for 10 min until charcoal settled and supernatant was filtered through 0.22 μm filter, as previously described [3].

Cells were seeded on coverslips in 24-well plates in antibiotic-free media 24 h before transfection. For the sake of specificity, pcDNA3.1(+) expression vector containing V5 tagged cDNA of GRα (pcDNA3.1-GRα-V5) were used for transfection, as previously reported [19,52], using FuGENE^®^ Transfection Reagent (#11988387001, Roche, Basel, Switzerland) and following the manufacturer’s instructions. GRβ intracellular localisation was assessed in our previous work with immunocytochemistry [19]. Twenty-four hours after transfection, for GR activation, GR agonist dexamethasone (100 nM, #D4902, Sigma-Aldrich, Merck, Kenilworth, NJ, USA) or dimethyl sulfoxide (DMSO, #276855, Sigma-Aldrich, Merck, Kenilworth, NJ, USA) as vehicle control was used for 4 h [3]. After a 4 h treatment, 100 nM MitoTracker Red (#M7512, Thermofisher Scientific, Waltham, MA, USA) was added for 30 min of incubation. Then, cells were washed twice with 1x phosphate saline buffer (PBS) and then fixed with 4% paraformaldehyde in PBS for 10 min. Cells were permeabilised in 0.2% TritonX-100 for 5 min, and then they were blocked in 2% bovine serum albumin in PBST at room temperature for 1 h. Then, cells were incubated with primary mouse monoclonal AntiV5 antibody in 1:1000 dilution (#R960-25, Invitrogen, Thermofisher Scientific, Waltham, MA, USA) overnight at 4 °C. FITC-conjugated donkey anti-mouse secondary antibody (#715-095-150, Jackson ImmunoResearch Ely, Cambridgeshire, UK) was applied in 1:200 dilution for 1 h at room temperature. Dako Fluorescent Mounting Medium (#S3023, Agilent Dako, Santa Clara, CA, USA) was used for covering before confocal laser scanning microscopy imaging (Bio-Rad MRC-1024 system, Bio-Rad, Richmond, CA, USA).

### 4.4. Statistical Methods

Based on the sample distribution determined by Shapiro-Wilks test and group number, unpaired T-test, Mann–Whitney U test, or Kruskall-Wallis ANOVA (with Dunn’s multiple comparison test) was used to identify statistical significance among different groups. Spearman’s correlation was applied and simple linear regression was used to identify the best-fit line. Kaplan-Meier analysis was used to assess the effect of PFS and OS following the cut-off determination by CutoffFinder [53]. For comparison of survival curves and the estimates of the hazard ratios, a log-rank test was run. Multivariate Cox regression analysis was used to investigate associations between survival and variables. Krippendorff’s alpha (ratio) was used to test inter-rater reliability and to investigate the degree of agreement among raters [46]. A *p*-value < 0.05 was considered statistically significant. The NCSS 2019 Statistical Software v. 19.0.9 (NCSS, LLC., Kaysville, UT, USA) was used for statistical analyses.

## 5. Conclusions

GRtotal and GRβ expression may have important relevance for patients with TNBC who receive supporting glucocorticoids (most frequently dexamethasone) as premedication to prevent hypersensitivity reactions to chemotherapy and as antiemetics.

Previous studies demonstrated that GR activation promotes disease progression in both cell line and patient-derived xenograft models [2]. It was also shown that even in the absence of the ligand, GR activation increases breast cancer cell migration in TNBC [3,4]. In addition, the GRβ isoform, which has an effect opposite to the main isoform GRα, plays an important role in breast cancer cell behaviour [19].

Based on our previous and published data, we aimed to take a step forward and validate the association of GR with clinical outcomes at the protein level using immunohistochemistry, a method routinely applied in pathology to characterise molecular subtypes through ER, PR, Her2, and Ki67 staining. Our data showed that GR can be detected in the cytoplasm of breast cancer cells and isoform-specific antibodies are crucial for detecting GR in breast cancer and understanding its clinicopathological associations, which may explain the conflicting literature findings.

We demonstrated an opposite association between GR and the proliferation index in the context of oestrogen receptor status. Based on this finding, survival analysis was performed on TNBC, where nuclear GRtotal and cytoplasmic GRβ showed opposing associations with patient outcomes. GRtotal was associated with a worse prognosis, while cytoplasmic localisation of GRβ correlated with a better prognosis in the multivariate analysis. These findings suggest that cellular localisation and GR isoforms may function differently in breast cancer, highlighting the multidirectional potential of GR. Our study is the first to evaluate the association of the GRβ isoform with clinicopathological parameters in breast cancer progression. Therefore, further investigations are needed before GR staining can be implemented into routine clinical practice.

## Figures and Tables

**Figure 1 ijms-25-11813-f001:**
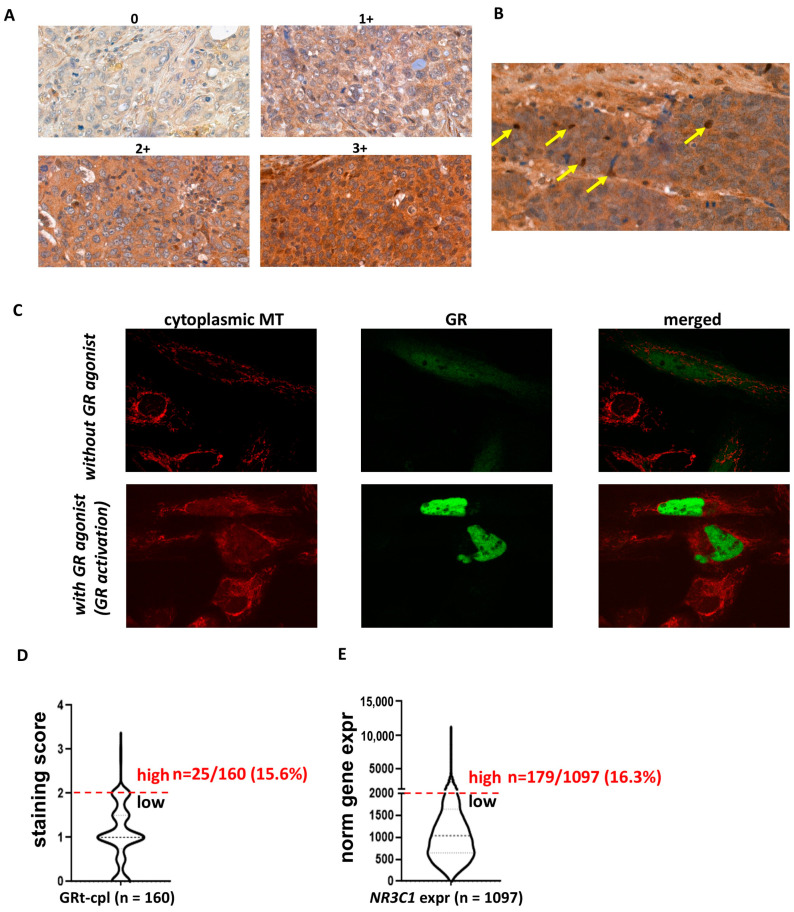
Cytoplasmic and nuclear staining patterns of GRtotal and GRβ in breast cancer. (**A**) Intensity categories used for quantifying cytoplasmic staining during IHC evaluation. (**B**) Representative images of nuclear staining, with positive nuclei indicated by yellow arrows. (**C**) Confocal fluorescent immunostaining illustrates the cytoplasmic localisation of the transfected GR in the absence of a GR ligand and nuclear localisation upon ligand binding. (**D**,**E**) The proportion of samples with high glucocorticoid receptor expression in both the current IHC study and the validation cohort (GRt-cpl: cytoplasmic GRtotal; GRβ-cpl: cytoplasmic GRβ; GRt-nucl: nuclear GRtotal; GRβ-nucl: nuclear GRβ; *NR3C1*: gene encoding GR; MT: mitotracker).

**Figure 2 ijms-25-11813-f002:**
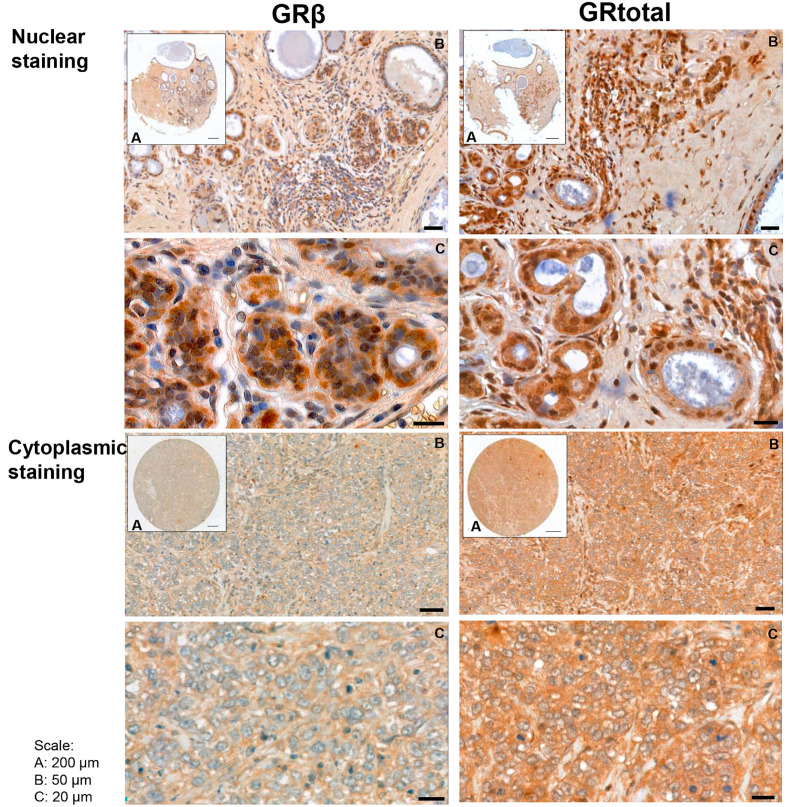
Representative images of GRtotal and GRβ staining in breast cancer. Mixed nuclear and cytoplasmic (above) and cytoplasmic only (below) staining. (A)–(C) panels indicate magnification. Black lines represent 200 μm on (A), 50 μm on (B), and 20 μm on (C) panels.

**Figure 3 ijms-25-11813-f003:**
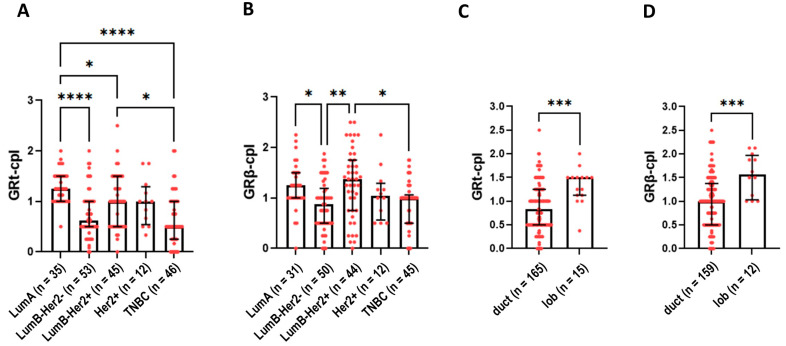
Cytoplasmic GRtotal (**A**) and GRβ staining (**B**) in intrinsic molecular subtypes. (**C**,**D**) panels present cytoplasmic GRtotal and GRβ staining in ductal and lobular breast cancer. Columns represent median ± interquartile range. *: *p* = 0.01–0.05; **: *p* = 0.001–0.01; ***: *p* = 0.0001–0.001; ****: *p* ≤ 0.0001. Abbreviations: GRt-cpl: cytoplasmic GRtotal; GRβ-cpl: cytoplasmic GRβ isoform; GRt-nucl: nuclear GRtotal; GRβ-nucl: nuclear GRβ isoform; Her2: receptor tyrosine-protein kinase erbB-2; duct: ductal breast cancer; lob: lobular breast cancer.

**Figure 4 ijms-25-11813-f004:**
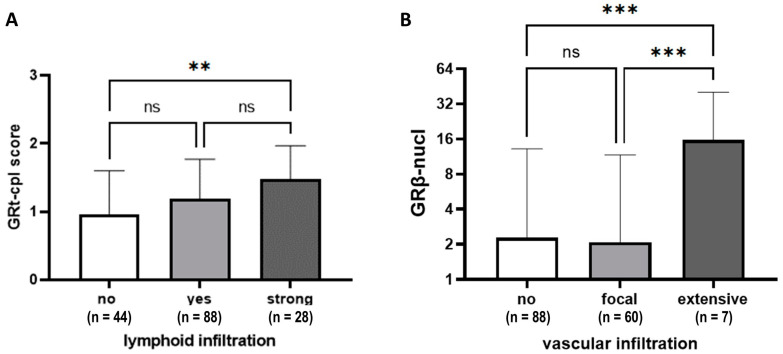
GRtotal and GRβ staining according to lymphoid (**A**) and vascular infiltration (**B**). Columns represent mean ± SD; **: *p* ≤ 0.01; ***: *p* ≤ 0.001; ns: not significant.

**Figure 5 ijms-25-11813-f005:**
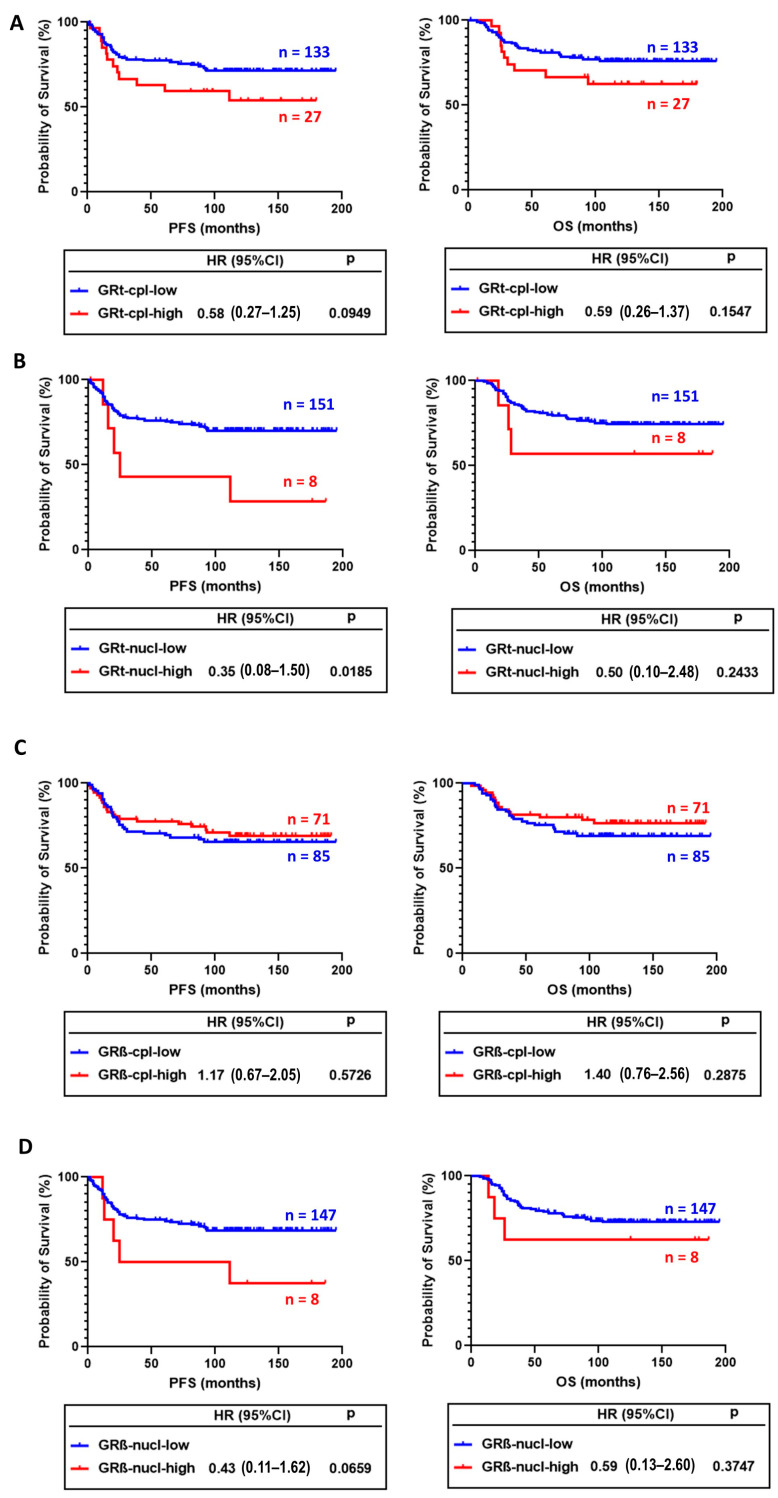
Association of cytoplasmic (**A**) and nuclear (**B**) GRtotal isoform staining with progression-free (PFS) and overall survival (OS). (**C**,**D**) panels present the association of cytoplasmic and nuclear GRβ staining with PFS and OS. Abbreviations: GRt-cpl: cytoplasmic GRtotal; GRβ-cpl: cytoplasmic GRβ isoform; GRt-nucl: nuclear GRtotal; GRβ-nucl: nuclear GRβ isoform; HR: hazard ratio.

**Table 1 ijms-25-11813-t001:** Characteristics of the two independent patient cohorts (subtype and prognostic groups) used for GR immunostaining.

	Subtype Cohort					Prognostic Cohort
Molecular Subtype	LumA	LumB-Her2-	LumB-Her2+	Her2+	TNBC	TNBC
ER	pos	pos	pos	neg	neg	neg
PR	pos	pos	pos/neg	neg	neg	neg
Her2+	neg	neg	pos	pos	neg	neg
Ki-67	<20%	≥ 20%	nr	nr	nr	nr
**patient #**	36	53	46	12	47	161
**patients’ age (years)**	60.9	58.5	58.8	55.6	57.4	55.7
(mean)						
(SD)	11.1	13.3	14.1	15.3	14.8	12.7
(range)	41–83	34–86	25–86	32–82	29–84	29–92

Abbreviations: LumA: luminal A subtype; LumB: luminal B subtype; TNBC: triple-negative breast cancer; ER: oestrogen receptor; PR: progesterone receptor; Her2: receptor tyrosine-protein kinase erbB-2; pos: positive; neg: negative, nr: not relevant for subtype classification; SD: standard deviation.

**Table 2 ijms-25-11813-t002:** Correlation among GR staining parameters and ER, PR, and Her2 status. Bold letters highlight statistical significance. Spearman R and *p* values are indicated in each comparison.

	ER	PR	Her2	GRt-cpl	GRt-nucl	GRβ-cpl	GRβ-nucl
**ER (n=196)**	**–**						
**PR (n = 196)**	**R = 0.72** ***p* = 6.37 × 10^−33^**	**–**					
**Her2 (n = 196)**	R = 0.02 *p* = 0.828	R = −0.08 *p* = 0.274	**–**				
**GRt-cpl (n = 193)**	**R = 0.24** ***p* = 0.001**	**R = 0.18** ***p* = 0.012**	R = 0.12 *p* = 0.103	**–**			
**GRt-nucl (n = 196)**	R = −0.13 *p* = 0.062	**R = −0.19** ***p* = 0.007**	R = 0.01 *p* = 0.885	**R = 0.36** ***p* = 2.48 × 10^−7^**	**–**		
**GRβ-cpl (n = 184)**	R = 0.13 *p* = 0.072	R = 0.07 *p* = 0.339	**R = 0.21** ***p* = 0.003**	**R = 0.41** ***p* = 1.52 × 10^−8^**	**R = 0.18** ***p* = 0.014**	**–**	
**GRβ-nucl (n = 186)**	R = −0.10 *p* = 0.162	R = −0.04 *p* = 0.555	R = −0.11 *p* = 0.120	R = 0.08 *p* = 0.261	**R = 0.20** ***p* = 0.007**	R = 0.04 *p* = 0.554	**–**

**Abbreviations:** GRt-cpl: cytoplasmic GRtotal; GRβ-cpl: cytoplasmic GRβ isoform; GRt-nucl: nuclear GRtotal; GRβ-nucl: nuclear GRβ isoform; ER: oestrogen receptor; PR: progesterone receptor; Her2: receptor tyrosine-protein kinase erbB-2.

**Table 3 ijms-25-11813-t003:** Correlation of cytoplasmic GRtotal and GRβ staining with Ki67 and mitotic count in luminal A and TNBC cohorts.

Breast Cancer Subtype (Sample Number)	GR Staining	Ki67 Spearman R (95%CI); *p* Value	Number of Mitoses Per 10 Fields Spearman R (95%CI); *p* Value
**LumA** **(n = 36)**	GRt-cpl	**−0.364 (−0.6280 to −0.02467); *p* = 0.032**	−0.258 (−0.5513 to 0.09299); *p* = 0.135
	GRβ-cpl	−0.270 (−0.5773 to 0.1040); *p* = 0.142	−0.223 (−0.5430 to 0.1531); *p* = 0.227
**TN** **(n = 161)**	GRt-cpl	**0.206 (0.02810 to 0.3718); *p* = 0.020**	**0.111 (0.06454 to 0.3685); *p* = 0.004**
	GRβ-cpl	**0.320 (0.1479 to 0.4734); *p* = 0.001**	**0.204 (0.04445 to 0.3546); *p* = 0.010**

Abbreviations: GRt-cpl: cytoplasmic GRtotal; GRβ-cpl: cytoplasmic GRβ isoform; GR: glucocorticoid receptor; LumA: luminal A subtype; TNBC: triple-negative subtype; Ki67: antigen Kiel 67.

**Table 4 ijms-25-11813-t004:** Prognostic factors associated with progression-free and overall survival in univariate analysis in TNBC. Significant results are indicated by bold letters.

Parameters		10-Year PFS	*p*		10-Year OS	*p*
		(95% CI)			(95% CI)	
age (years)	<62	74% (65–82)	**0.044**		79% (71–87)	**0.025**
	≥62	58% (45–71)			63% (50–76)	
type of surgery *	1	77% (68–85)	**0.001**		81% (73–88)	**0.002**
	2	51% (38–65)			58% (45–72)	
vascular invasion	no	79% (70–87)	**<0.001**		85% (77–93)	**<0.001**
	yes	53% (41–66)			57% (46–69)	
perinodal invasion	no	61% (44–78)	**0.034**		65% (49–81)	**0.028**
	yes	36% (17–55)			35% (16–54)	
surgical margin (mm)	<1	46% (26–65)	**0.003**	<0.7	60% (41–79)	0.061
	≥1	72% (65–80)		≥0.7	76% (68–83)	
Nottingham prognostic index	<3.58	82% (74–90)	**<0.001**	<3.6	88% (81–95)	**<0.001**
	≥3.58	49% (37–61)		≥3.6	53% (41–65)	
tumour size (mm)	<29	75% (66–83)	**<0.001**	<30	80% (73–87)	**<0.001**
	≥29	49% (34–64)		≥30	53% (37–68)	
pN	0	78% (70–87)	**<0.001**		85% (78–93)	**<0.001**
	1	51% (38–64)			53% (41–66)	
positive/removed lymph node ratio	<0.087	77% (70–85)	**<0.001**	<0.076	84% (77–91)	**<0.001**
	≥0.087	42% (27–58)		≥0.076	44% (30–59)	
stage	1–2	75% (68–83)	**<0.001**	1–2a	83% (75–90)	**<0.001**
	3	28% (9–46)		2b-3	52% (37–66)	
neutrophil count (10^9^/l)	<2.89	36% (11–61)	**0.01**	<7.29	66% (56–75)	0.064
	≥2.89	67% (58–77)		≥7.29	93% (79–100)	
SII (10^9^/l)	<354	31% (4–59)	**0.019**		65% (56–75)	0.068
	≥354	67% (58–76)			89% (74–100)	
BMI	<29.8	63% (55–72)	**0.035**		65% (53–77)	0.086
	≥29.8	84% (70–97)			78%/ (70–87)	
adjuvant treatment	no or RT or ant	45% (29–61)	**<0.001**		53% (38–69)	**<0.001**
	tax + ant ± RT	68% (57–80)			73% (62–84)	
	CMF ± RT or ant + RT	83% (73–93)			87% (78–96)	
p53	0	40% (0–83)	**0.045**		40% (0–83)	**0.021**
	>0	90% (71–100)			90% (71–100)	
DCIS grade	1–2 (or no DCIS)	63% (51–75)	0.08		65% (53–77)	**0.035**
	3 (or necrosis)	72% (62–81)			79% (70–87)	
body mass (kg)	<72	64% (54–73)	0.117		67% (57–77)	**0.04**
	≥72	75% (64–87)			82% (73–91)	
GRt-nuc	GRt-nucl-low (<65)	81% (73–90)	**0.0185**		85% (77–93)	0.2433
	GRt-nucl-high (≥65)	17% (−24–60)			60% (29–92)	
GRß-nuc	GRß-nucl-low (<30)	81% (72–90)	0.0659		87% (79–94)	0.3747
	GRß-nucl-high (≥30)	20% (−24–65)			18% (2–34)	
GRt-cpl	GRt-cpl-low (<2)	82% (72–91)	0.0949		87% (78–95)	0.1547
	GRt-cpl-high (≥2)	32% (11–52)			48% (28–67)	
GRß-cpl	GRß-cpl-low (<0.83)	65% (54–76)	0.5726		101% (86–116)	0.2875
	GRß-cpl-high (≥0.83)	81% (67–95)			83% (74–92)	
Ki-67	<75	74% (65–83)	0.06		67% (55–79)	0.099
	≥75	37% (38–75)			81% (72–91)	
body height (cm)	<167	72% (64–81)	0.062		77% (69–85)	0.089
	≥167	56% (41–71)			62% (48–77)	

** 1 excision/ablation + sentinel lymph node biopsy, 2 excision/ablation ± axillary block dissection. **Abbreviations:** ant: anthracyclines; CI: confidence interval; CMF: cyclophosphamide/methotrexate/5-fluorouracil; DCIS: ductal carcinoma *in situ*; GRt-cpl: cytoplasmic GRtotal; GRβ-cpl: cytoplasmic GRβ isoform; GRt-nucl: nuclear GRtotal; GRβ-nucl: nuclear GRβ isoform; HR: hazard ratio; nuc: nuclear; IDC: invasive ductal carcinoma; NLRRT: radiotherapy; PFS: progression-free survival; OS: overall survival; SII: systemic immune-inflammation index = platelet*neutrophil/lymphocyte, tax: taxanes.*

**Table 5 ijms-25-11813-t005:** Prognostic factors associated with overall survival in multivariate Cox analysis. Significant results are indicated by bold letters.

Parameters		10-Year OS HR (95% CI)	*p*
Age (years)	<62	1 (ref.)	0.116
	≥62	1.81 (0.86–3.79)	
Positive/removed lymph node ratio	<0.076	1 (ref.)	***p* < 0.01**
	≥0.076	6.66 (2.93–15.1)	
SII (10^9^/l)	<354	1 (ref.)	0.345
	≥354	0.58 (0.19–1.81)	
BMI	<29.8	1 (ref.)	0.677
	≥29.8	0.79 (0.26–2.42)	
DCIS grade	1–2 (or DCIS < 10%)	1 (ref.)	**0.017**
	3 (or necrosis)	0.41 (0.19–0.85)	
Body weight (kg)	<72	1 (ref.)	0.149
	≥72	1.73 (0.82–3.63)	
GRß-cpl	low	1 (ref.)	**0.036**
	high	0.46 (0.22–0.95)	

*BMI: body mass index; DCIS: ductal carcinoma *in situ*; GRβ-cpl: cytoplasmic GRβ isoform; HR: hazard ratio; OS: overall survival; SII: systemic immune-inflammation index = platelet*neutrophil/lymphocyte.*

## Data Availability

All data generated or analysed during this study are included in this article and its Appendix A. The clinical data analysed during the current study are available from the corresponding author on reasonable request. RNA-seq data were retrieved from the Genotype-Tissue Expression (GTEx) portal (version no. 7–15 May 2019) through TNMplot.com as indicated in the Materials and Methods Section.

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
