# Peer review of "Glucocorticoid Receptor Isoforms in Breast Cancer Raise Implications for Personalised Supportive Therapies"

_ijms, 2024, doi:10.3390/ijms252111813_

Round 1
Reviewer 1 Report
Comments and Suggestions for Authors
Attached

Extensive editing of English language required.
Author Response
Response to Reviewer 1:
We thank the reviewer for his/her time and thoughtful feedback on our manuscript. Please find our responses to each of the points raised below.
Reviewer Response Manuscript no.: ijms-3232238
Title: Glucocorticoid receptor isoforms in breast cancer raise implications for personalized supportive therapies
This is a prospective study to investigate Glucocorticoid receptor and its isoform expression and assess their associations with clinicopathological findings and prognosis in breast cancer. Kindly find my comments:
- GR has multidirectional potential and different cellular localization and GR isoforms can function differently in breast cancer, which should be extensively studied as basic research before studying its association with clinicopathological characteristics and clinical outcome; and before recommending its implementation in clinical practice as personalized treatment strategies.
Response: We agree with the Reviewer that GR has complex, context-dependent functions, and therefore, extensive preclinical research should be conducted before its application in clinical practice. At the same time, numerous studies have investigated the role of GR in breast cancer in preclinical models. For instance, Obradovic et al. (2019) demonstrated that GR activation promotes disease progression in both cell line and patient-derived xenograft models. West et al. (2016) and Tonsing-Carter et al. (2019) performed extensive studies on ER-GR interactions in oestrogen receptor-positive breast cancer to examine GR action in the presence of ER. Additionally, Perez-Kerkvilet et al. (2020) and Dwyer et al. (2023) showed that even in the absence of the ligand, GR activation increases breast cancer cell migration in triple-negative breast cancer.
We previously published, for the first time in breast cancer research, that the GRβ isoform, which has an opposite effect to the main isoform GRα, plays an important role in breast cancer cell behaviour (Butz et al. 2023). Pan et al. (2011) demonstrated, there is an opposite association between GR expression levels and survival in ER-positive and ER-negative breast cancers. However, no subsequent studies have confirmed their findings at the protein level.
Based on our previous data and previously published data, we aimed to take a step forward and validate the association of GR expression at the protein level using immunohistochemistry with outcomes. We applied this routinely used method in breast cancer to characterize molecular subtypes through ER, PR, Her2, and Ki67 staining. Since RNA level measurements are not part of routine pathology and RNA and protein levels do not necessarily correlate, our study brings GR a step closer to clinical relevance. Additionally, in this study, we were the first to demonstrate the association between GRβ and better survival in TNBC patients, which raises the possibility that patients with high GRβ expression may not be at risk of increased disease progression upon glucocorticoid treatment.
In agreement with the Reviewer, we also emphasized in the revised manuscript in the conclusion section that before implementing GR immunohistochemistry into routine clinical practice, further investigations are needed.
Please find this clarification highlighted in the conclusion section (lines 390-411).
Obradović, M.M.S.; Hamelin, B.; Manevski, N.; Couto, J.P.; Sethi, A.; Coissieux, M.-M.; Münst, S.; Okamoto, R.; Kohler, H.; Schmidt, A.; et al. Glucocorticoids Promote Breast Cancer Metastasis. Nature 2019, 567, 540–544, doi:10.1038/s41586-019-1019-4
West, D.C.; Pan, D.; Tonsing-Carter, E.Y.; Hernandez, K.M.; Pierce, C.F.; Styke, S.C.; Bowie, K.R.; Garcia, T.I.; Kocherginsky, M.; Conzen, S.D. GR and ER Coactivation Alters the Expression of Differentiation Genes and Associates with Improved ER+ Breast Cancer Outcome. Mol. Cancer Res. MCR 2016, 14, 707–719, doi:10.1158/1541-7786.MCR-15-0433.
Tonsing-Carter, E.; Hernandez, K.M.; Kim, C.R.; Harkless, R.V.; Oh, A.; Bowie, K.R.; West-Szymanski, D.C.; Betancourt-Ponce, M.A.; Green, B.D.; Lastra, R.R.; et al. Glucocorticoid Receptor Modulation Decreases ER-Positive Breast Cancer Cell Proliferation and Suppresses Wild-Type and Mutant ER Chromatin Association. Breast Cancer Res. BCR 2019, 21, 82, doi:10.1186/s13058-019-1164-6.
Perez Kerkvliet, C.; Dwyer, A.R.; Diep, C.H.; Oakley, R.H.; Liddle, C.; Cidlowski, J.A.; Lange, C.A. Glucocorticoid Receptors Are Required Effectors of TGFβ1-Induced P38 MAPK Signaling to Advanced Cancer Phenotypes in Triple-Negative Breast Cancer. Breast Cancer Res. 2020, 22, 39, doi:10.1186/s13058-020-01277-8.
Dwyer, A.R.; Perez Kerkvliet, C.; Truong, T.H.; Hagen, K.M.; Krutilina, R.I.; Parke, D.N.; Oakley, R.H.; Liddle, C.; Cidlowski, J.A.; Seagroves, T.N.; et al. Glucocorticoid Receptors Drive Breast Cancer Cell Migration and Metabolic Reprogramming via PDK4. Endocrinology 2023, 164, bqad083, doi:10.1210/endocr/bqad083
Butz, H.; SaskÅ‘i, É.; Krokker, L.; Vereczki, V.; Alpár, A.; Likó, I.; Tóth, E.; SzÅ‘cs, E.; Cserepes, M.; Nagy, K.; et al. Context-Dependent Role of Glucocorticoid Receptor Alpha and Beta in Breast Cancer Cell Behaviour. Cells 2023, 12, 784, doi:10.3390/cells12050784.
Pan, D.; Kocherginsky, M.; Conzen, S.D. Activation of the Glucocorticoid Receptor Is Associated with Poor Prognosis in Estrogen Receptor-Negative Breast Cancer. Cancer Res. 2011, 71, 6360–6370, doi:10.1158/0008-5472.CAN-11-0362
- The aim of the study is not very clear and is not consistent with the conclusion, where the authors stated in the abstract that „GR activation may promote metastasis in estrogen receptor-negative breast cancer. However, the role of GRβ isoform , which has opposing effects to the main isoform, has not been studied in clinical samples”, however in the conclusion section they focused only on TNBC.
Response: We thank the Reviewer to point to this issue.
In the current study we formulated two aims: 1) to analyse the intracellular localization of total GR and GRβ in vitro using plasmid constructs and fluorescent immunocytochemistry; 2) to perform immunostaining for total GR and GRβ on two breast cancer cohorts: (1) 194 clinical breast cancer samples to compare different molecular subtypes, and (2) 161 triple-negative breast cancer (TNBC) samples to analyse the effect of GR on survival. We clarified this in the abstract (lines 25-28) and in the introduction section (lines 90-97), please find it highlighted. We validated GR expression distribution using RNA data from 1,097 TNBC cases (Figure 1E and line 138-142).
Indeed, since GR protein expression was associated with higher Ki67 in TNBC and with lower Ki67 in luminal A breast cancer, we focused our survival analysis on TNBC in the subsequent investigation. We clarified this in the revised version of the abstract (lines 27-28 and 38-42) and we added the relevance of the GR staining in luminal A and TNBC to the conclusion section as well to make the aims in line with conclusion (line 402-405). Please find highlighted.
- Although the authors stated in the abstract that they analyzed the intracellular localization immunocytochemistry, as well as in 355 clinical breast cancer samples, supplemented by 25 data from 1,097 triple-negative breast cancer (TNBC) cases, however the this is not reflected in the results.
Response: In agreement with the Reviewer comment we clarified the text according to the following: Different expression and cellular localization of GRtotal and GRβ were observed in the context of molecular subtypes, underscoring the complex role of GR in breast cancer. An inverse association between cytoplasmic GRtotal and the Ki67 proliferation index was observed in luminal A and TNBC. Regarding the impact of GR on outcomes in TNBC patients, while cytoplasmic GRβ was associated with a better prognosis, patients with nuclear GRtotal staining may be at a higher risk of disease progression, as it negatively affects survival. Please find highlighted on lines 37-42 in the abstract
We present the staining of GRtotal and GRβ in the Results section as GR positive vs negative samples: (lines 128-132). We also presented differences in positive staining strength (weak-moderate, and strong on lines 134-138; while findings on the 1097 TNBC validation cohort on lines 138-142 and Figure 1D-E.
- The manuscript should be extensively revised for English editing and formatting.
Response: We thank the Reviewer for his/her remark we performed an extensive language check on the manuscript.
- All abbreviations should be extensively revised throughout the manuscript and full names stated ONLY first time they appear in the text. For example, TNBC, IHC, OS, PFS, LI
Response: We revised the abbreviations so that full names are now included only at their first appearance. The only exceptions are in the figure and table legends, where abbreviations are provided for each item, as each figure and table should stand alone according to the publication guidelines.
- Results section (line 87) the paragraph should be removed „This section may be divided by subheadings. It should provide a concise and precise description of the experimental results, their interpretation, as well as the experimental conclusions that can be drawn”
Response: We thank the Reviewer for his/her accurate remark, we deleted the abovementioned sentence that remained from the template given by the journal. We apologize for this mistake.
- Results need to be more adequately described.
Response: In response to the Reviewer's suggestion, we have rephrased the results section to emphasize the following points:
In paragraph 2.1, we first demonstrated the validity of cytoplasmic GR staining using confocal immunofluorescence images following in vitro transfection of the GR and ligand activation upon dexamethasone treatment. We then described the GR staining patterns observed in clinical samples (both cytoplasmic and nuclear), noting the distribution of both GR-positive and GR-negative cases. Among the GR-positive samples, we assessed the intensity of GR-total and GRβ staining in terms of their nuclear and cytoplasmic localization.
In paragraph 2.2, we characterized GRtotal and GRβ staining across different molecular and histological subtypes of breast cancer.
Finally, in paragraph 2.3, building on our results from section 2.2 and literature data highlighting the tumour-promoting effect of GR activation in TNBC, we focused exclusively on TNBC, describing the correlation between GR-total and GRβ staining and clinicopathological parameters, including survival.
Please find highlighted the clarifications throughout the Results section.
- The figure ligands should clearly describe the figure and all abbreviations in the figure written down in the ligand (Figure 1).
In response to the Reviewer’s request, we have improved the figure legends, please find highlighted throughout the whole manuscript. Following the journal’s style guidelines, we have included abbreviations after the figure legends above the figures and after the tables, as appropriate.
- Table 1, 4, 5 should not be divided in separate pages.
Response: We adjusted and reformatted the tables in the revised manuscript so they are not split across different pages. Only Table 4 is presented on two pages, due to its size.
- Figure 3 (line 154): font in the figure is very small. Panels A & B should be separated, and sub-numbering given to the figures.
Response: In agreement with the reviewer, we doubled the size of Figure 3 and separated it into four different panels, each with a corresponding legend.
- Figure 4 (line191): very difficult to read. Panels should be enlarged with better resolution and clear description of the figure should be written in the ligand
Response: As per the reviewer’s request, we split Figure 4 into two separate figures (Figure 4 and Figure 5) to improve readability. Additionally, we increased the size of Figure 5 and arranged the graphs into four different panels with corresponding legends.

Reviewer 2 Report
Comments and Suggestions for Authors
In this manuscript, the authors have investigated clinical breast cancer samples in order to clarify the role of glucocorticoid receptors in breast cancer metastasis. The results of the study indicate that there is a relationship between GRbeta expression and prognosis, and the authors conclude that the use of glucocorticoids may have a negative impact on the survival of breast cancer patients. While the data in this paper may contribute to the treatment of breast cancer, there are some points that need to be improved.
<Major>
1. In section 4.3, there is a description of cell culture, but there is no description of what kind of cells were used. Presumably, the cells were taken from the tissue of breast cancer patients, but if that is the case, shouldn't the process be described?
2. There are descriptions of Figure 1D-E in the main text, but there are no corresponding figures.
Author Response
Response to Reviewer 2:
We thank the reviewer for his/her time and thoughtful feedback on our manuscript. Please find our responses to each of the points raised below.
Reviewer 2
Comments and Suggestions for Authors
In this manuscript, the authors have investigated clinical breast cancer samples in order to clarify the role of glucocorticoid receptors in breast cancer metastasis. The results of the study indicate that there is a relationship between GRbeta expression and prognosis, and the authors conclude that the use of glucocorticoids may have a negative impact on the survival of breast cancer patients. While the data in this paper may contribute to the treatment of breast cancer, there are some points that need to be improved.
Response: We thank the reviewer for his/her time and thoughtful feedback on our manuscript.
<Major>
- In section 4.3, there is a description of cell culture, but there is no description of what kind of cells were used. Presumably, the cells were taken from the tissue of breast cancer patients, but if that is the case, shouldn't the process be described?
Response: In this study, we used HeLa cells which is recognized as the most commonly used cell line for investigating human cellular and molecular biology and has been reported as a relevant model for studying the glucocorticoid receptor due to its epithelial origin (references provided below). Using confocal fluorescent imaging we demonstrated GR cytoplasmic localization and confirmed that GR translocates into the nucleus in the presence of a ligand (dexamethasone, a GR agonist). This proves that the observed cytoplasmic staining with the antibodies used is specific and a valid finding. In our previous study (Butz et al. 2023) we already used MDA-MB231 and HS578T breast cancer cells for GR transfections. In this study, however, we implemented confocal microscopy to scrutinize GR cellular localization. The detailed description of the cells, transfection and treatment can be found in the Methods section (line 333-369) of the revised manuscript.
Vrzal, R.; Ulrichová, J.; Dvorák, Z.; Pávek, P. Glucocorticoid Receptor Functions in HeLa Cells Are Perturbed by 2,3,8,9-Tetrachlorodibenzo-p-Dioxin (TCDD). Drug Metab. Lett. 2007, 1, 311–314.
Shimojo, M.; Hiroi, N.; Yakushiji, F.; Ueshiba, H.; Yamaguchi, N.; Miyachi, Y. Differences in Down-Regulation of Glucocorticoid Receptor MRNA by Cortisol, Prednisolone and Dexamethasone in HeLa Cells. Endocr. J. 1995, 42, 629–636.
Gupta, V.; Wagner, B.J. Expression of the Functional Glucocorticoid Receptor in Mouse and Human Lens Epithelial Cells. Investig. Ophthalmol. Vis. Sci. 2003, 44, 2041–2046.
DvoÅ™ák, Z.; Vrzal, R.; Maurel, P.; Ulrichová, J. Differential Effects of Selected Natural Compounds with Anti-Inflammatory Activity on the Glucocorticoid Receptor and NF-ΚB in HeLa Cells. Chem. Biol. Interact. 2006, 159, 117–128.
Molina, M.L.; Guerrero, J.; Cidlowski, J.A.; Gatica, H.; Goecke, A. LPS Regulates the Expression of Glucocorticoid Receptor α and β Isoforms and Induces a Selective Glucocorticoid Resistance in Vitro. J. Inflamm. 2017, 14, 22.
Kino, T.; Manoli, I.; Kelkar, S.; Wang, Y.; Su, Y.A.; Chrousos, G.P. Glucocorticoid Receptor (GR) Beta Has Intrinsic, GRalpha-Independent Transcriptional Activity. Biochem. Biophys. Res. Commun. 2009, 381, 671–675.
Butz H, SaskÅ‘i É, Krokker L, Vereczki V, Alpár A, Likó I, Tóth E, SzÅ‘cs E, Cserepes M, Nagy K, Kacskovics I, Patócs A. Context-Dependent Role of Glucocorticoid Receptor Alpha and Beta in Breast Cancer Cell Behaviour. Cells. 2023 Mar 1;12(5):784.
- There are descriptions of Figure 1D-E in the main text, but there are no corresponding figures.
Response: We thank the Reviewer for pointing out this mistake, and we have corrected Figure 1. We greatly appreciate the Reviewer's thorough and accurate work.
